# A Laminated Spherical Tsunami Shelter with an Elastic Buffer Layer and Its Integrated Bulge Processing Method

Junfu Hou [1], Li Chen [1], Jingchao Guan [1], Wei Zhao [2], Ichirou Hagiwara [3] and Xilu Zhao [1,*]

[1] Department of Mechanical Engineering, Saitama Institute of Technology, Saitama 369-0293, Japan; kevinisperfect0809@gmail.com (J.H.); chenli98721@gmail.com (L.C.); guanjingchao123@gmail.com (J.G.)
[2] Space C5 Co., Ltd., Tokyo 101-8301, Japan; shunsuke0390@gmail.com
[3] Organization for the Strategic Coordination of Research and Intellectual Properties, Meiji University, Tokyo 101-8301, Japan; ihagi@meiji.ac.jp
[*] Correspondence: zhaoxilu@sit.ac.jp

**Abstract:** When a tsunami occurs, people can enter floating shelters and save their lives. Tsunami shelters consisting of thin-walled fiber-reinforced plastic (FRP) spherical shells have been developed and are currently in use. In this study, a novel three-layer laminated spherical tsunami shelter and its fabrication method have been proposed as an alternative to the conventional thin-walled spherical FRP tsunami shelter. First, the inner and outer layers were made of thin-walled stainless-steel spherical shells using the integral hydro-bulge-forming (IHBF) method. The inter-layers between the inner and outer layers were filled with elastic rubber to provide a laminated spherical tsunami shelter with elastic cushioning layers. After the fabrication process was developed, a laminated spherical tsunami shelter with a plate thickness of 1.0 mm, an inner spherical shell design radius of 180 mm, and an outer spherical shell design radius of 410 mm was fabricated. The shape accuracy of the process was determined. The roundness values of the inner and outer layers of the spherical shell were 0.88 and 0.85 mm, respectively. The measured radii of the actual inner and outer spherical shells were 180.50 and 209.97 mm, respectively, and the errors between the design and measured radii were 0.28% and −0.01%. In this study, acceleration sensors were attached to the inner and outer layers of the processed, laminated spherical tsunami shelter. A hammer impact load was applied to the outer layer, and the response acceleration values measured by the acceleration sensors in the inner and outer layers were compared. It was confirmed that the response acceleration value of the inner layer was 10.17% smaller than that of the outer layer. It was then verified that the spherical tsunami shelter proposed in this study has a good cushioning effect and processing performance.

**Keywords:** spherical tsunami shelter; laminated spherical shell; integrated bulge processing method; elastic buffer laminated shell; plastic forming of thin steel plate

## 1. Introduction

Disasters such as earthquakes and tsunamis occur naturally, and people suffer extensive damage due to them. Therefore, floating tsunami shelters have been developed to save lives from tsunamis [1,2]. When a tsunami occurs, people can enter floating tsunami shelters to save their lives.

To develop a floating tsunami shelter, it is important to investigate the distribution of flow velocity and wave force when a tsunami approaches land; thus, the wave pressure acting on the structure by the tsunami must be studied [3–5].

Floating tsunami shelters are broadly classified into two types. One is those that are bound to land shores with chains or vertical guide links; thus, when a tsunami occurs, they will float and sink within a certain range of the land [6–8]. The other type is those that are independent and float on the sea surface; thus, when a tsunami occurs, they will move from the land's coast and float with the waves [9–11].

Floating tsunami shelters that are independent from the coast have relatively simple structures and low costs. Additionally, spherical tsunami shelters are relatively easy to use in terms of receiving random wave forces from tsunamis. Therefore, floating spherical tsunami shelters have been studied with regards to various aspects, such as mechanical properties [12–15], and research is being conducted on cubic polyhedral and origami structures, providing basic research results that are useful for the design and manufacture of tsunami shelters composed of thin-walled shells [16–19].

Actual commercial products have been developed [20]. Current spherical tsunami shelters are made of single-layer, thin-walled FRP composite materials; therefore, they are not sufficiently strong against rocks and other objects, and the noise caused by waves hitting them can make people feel uncomfortable. The nonlinear dynamics and stability problems of thin spherical shells under external pressure have been investigated, and the problems of single-layer shells have been investigated from a theoretical angle [21]. To address these problems, it is desirable to develop laminated spherical tsunami shelter structures with elastic buffer layers instead of single layers. The dynamic responses of laminated spherical shells with foam material when subjected to explosions and impact loads has been investigated, and the cushioning effects of laminated spherical shells against impact loads has been verified [22].

In addition, prior to developing a laminated spherical tsunami shelter structure, it is necessary to consider the high-precision processing of spherical, thin-metal ball shells. However, it is difficult to process general thin-walled spherical metal containers with high precision [23,24]. To solve this problem, an integral hydro-bulge-forming method (IHBF) has been proposed [25–27]. However, it is still difficult to fabricate laminated spherical tsunami shelter structures using the elastic buffer layer proposed in the studies based on the IHBF method.

In this study, a novel method was developed to fabricate a laminated spherical tsunami shelter structure composed of three-layered shells. The inner and outer layers were made of thin stainless-steel spherical shells using the IHBF method. By filling the intermediate layer between the inner and outer layers with silicone rubber, a laminated spherical tsunami shelter structure in which the inner, outer, and intermediate layers were closely bonded was obtained. First, the processing method for the proposed laminated spherical tsunami shelter was developed and a specific processing flowchart was created. Subsequently, the roundness values of the inner and outer layers of the ball shell of the laminated spherical tsunami shelter were measured to verify the shape accuracy of the proposed processing method. Furthermore, to put the proposed processing method into practical use, we derived calculation formulas for the dimensions of the inner and outer spherical shells and for the internal water pressure required for bulge formation. Finally, an impact load was applied to the processed laminated spherical tsunami shelter using a hammer to verify the cushioning effect.

## 2. Materials and Methods

### 2.1. Single-Layer Spherical Tsunami Shelter Structure

To avoid tsunami damage, a spherical tsunami shelter, as shown in Figure 1, has been developed and used. When a tsunami occurs, people can enter the spherical tsunami shelter, as shown in Figure 1, and the shelter can be moved by waves.

It is difficult to develop laminated spherical tsunami shelter structures with elastic buffer layers to overcome the shortcomings of current single-layer spherical FRP tsunami shelters.

As the first step in developing a laminated spherical tsunami shelter structure, it is necessary to investigate a high-precision manufacturing method for thin metallic spherical shells.

Conventionally, the main method for processing thin, spherical metal shells involves two steps. First, a thin curved steel plate is processed using the sheet metal pressing method with a mold. Subsequently, a spherical shell is obtained by sequentially welding the curved

steel plate parts. This manufacturing method not only requires processing time and high costs but also makes it difficult to ensure the shape accuracy of the spherical shell.

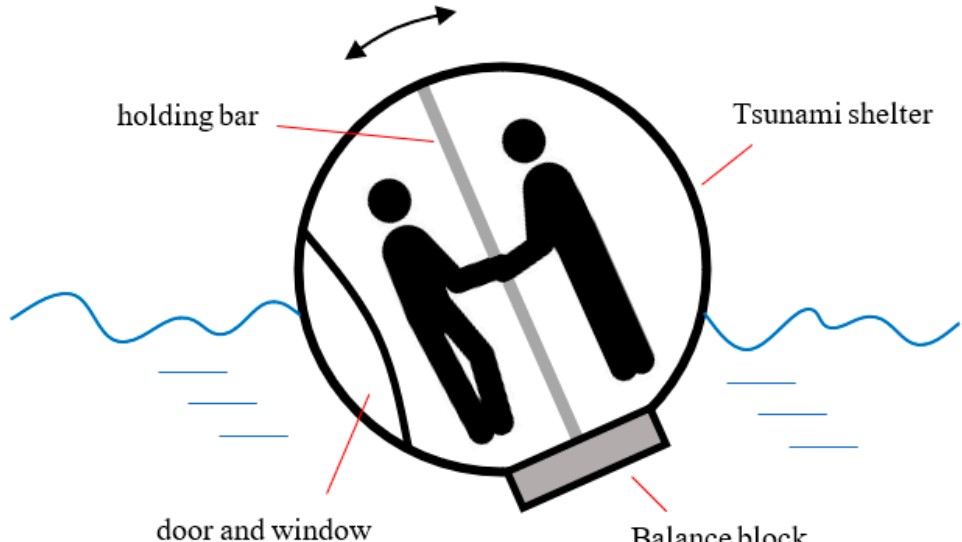

**Figure 1.** Spherical tsunami shelter currently developed and used [20].

Therefore, as shown in Figure 2, an IHBF method for processing spherical containers has been proposed and studied [23–27]. First, a laser processing machine is used to cut the parts from a flat metal plate. Each part is then folded and welded along its edges to create a closed, preformed shell. Finally, a spherical container is obtained by applying water pressure to the sealed-closed, preformed shell and elastically forming it from the inside through the expansion force of the water pressure.

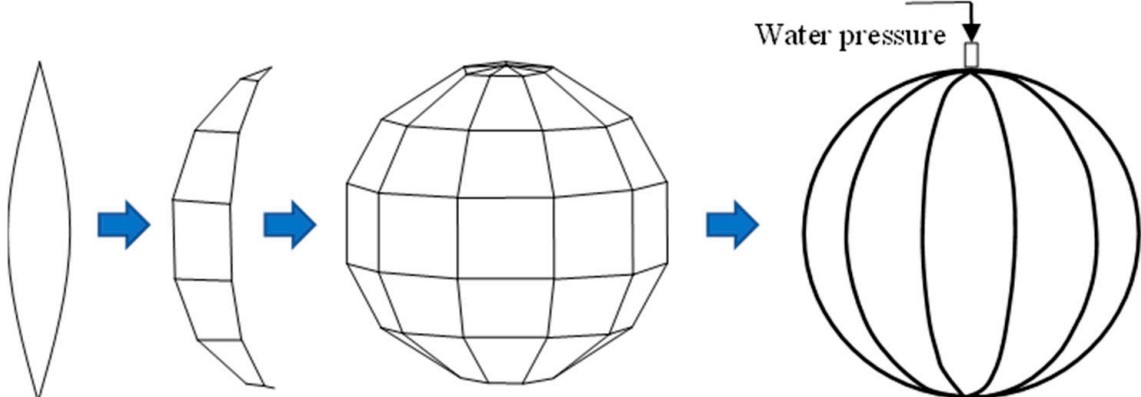

**Figure 2.** Spherical shell manufacturing using the integral hydro-bulge-forming method (IHBF) [23].

When processing a thin spherical shell using the IHBF method, as shown in Figure 2, the shape accuracy of the formed spherical shell is significantly affected by the shape of the closed, preformed shell. It is advantageous to design the closed, preformed shell with a spherical, symmetrical shape, such as a regular polyhedron.

However, there are only five types of regular polyhedron that have symmetry that is close enough to three-dimensional spherical symmetry: tetrahedron, hexahedron, octahedron, dodecahedron, and icosahedron, as shown in Figure 3 [28].

Among regular polyhedrons, the icosahedron, which has the largest number of faces, was the subject of this study. As shown in Figure 3e, a regular icosahedron consists of 20 equilateral triangles and 12 vertices.

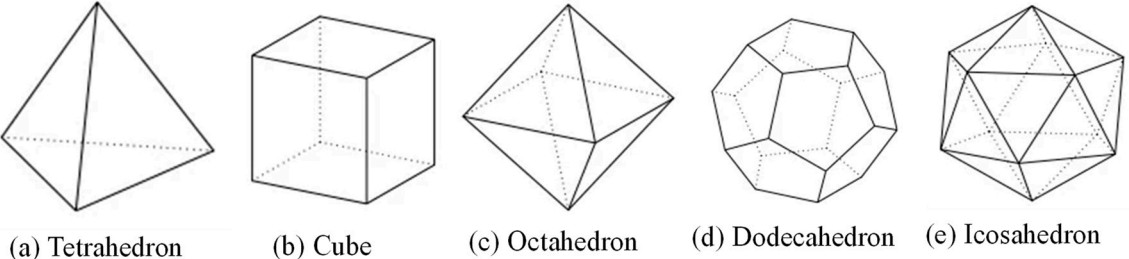

(a) Tetrahedron  (b) Cube  (c) Octahedron  (d) Dodecahedron  (e) Icosahedron

**Figure 3.** Regular polyhedrons closest to five types of spherical symmetry [28].

Here, the sides of a regular icosahedron were divided into three equal parts, and the vertices were cut through the trisecting points near each vertex, as indicated by the dotted lines in Figure 4a. A soccer-ball-shaped polyhedron was obtained, as shown in Figure 4b. The part enclosed by the dotted line in Figure 4a corresponds to the regular black pentagon in Figure 4b, and the part indicated by the hatching in Figure 4a corresponds to the regular hexagon in Figure 4b.

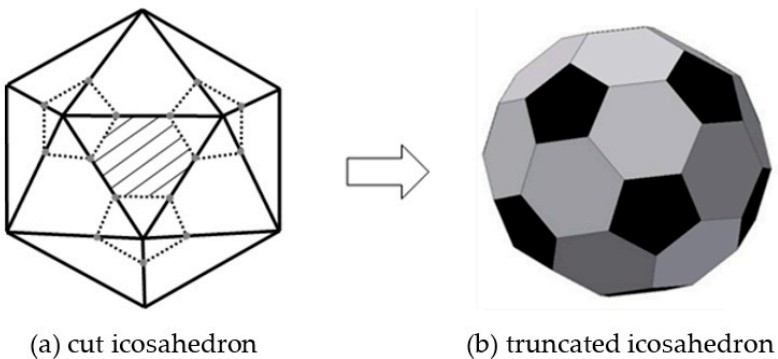

(a) cut icosahedron  (b) truncated icosahedron

**Figure 4.** Regular icosahedron and soccer-ball-type polygonal box.

For examination, one regular hexagon of the soccer-ball-shaped polyhedron shown in Figure 4b was extracted and is shown in Figure 5, where $o$ is the center point of the soccer-ball-shaped polyhedron, $o_6$ is the center point of the regular hexagonal flat plate, $a$ is the side length of the hexagonal flat plate, and $R$ is the distance from the center point to the vertex of the soccer-ball-shaped polyhedron. $r_6$ is the distance from the center point to the center point of the regular hexagonal plate.

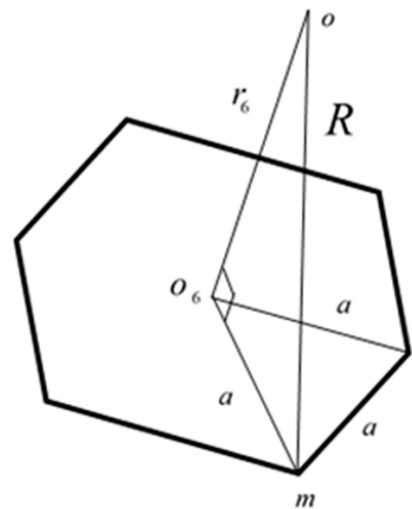

**Figure 5.** Parameters of the regular hexagonal flat plate.

The distance from the center point of the soccer-ball-shaped polyhedron to the center point of the regular hexagonal flat plate was calculated using the following equation [22]:

$$r_6 = \frac{3\sqrt{3} + \sqrt{15}}{4} a \tag{1}$$

Considering the right-angled triangle, $oO_6m$, shown in Figure 5, the following relations were obtained:

$$a = \sqrt{R^2 - r_6^2} = 0.40355R \approx 0.4R \tag{2}$$

When the soccer-ball-shaped polyhedron was formed as a preformed spherical shell using the IHBF method, the partial plastic formations of straight weld lines into curves and flat plate parts into curved surfaces became dominant. Therefore, it can be considered that the distance, $R$, from the center point to the vertex of the soccer-ball-shaped polyhedron before the formation was equal to the radius of the spherical shell after formation.

### 2.2. Laminated Spherical Tsunami Shelter Structure with Elastic Buffer Layer

A new processing method was developed based on the thin soccer-ball-shaped spherical structure described in the previous section.

A laminated spherical tsunami shelter body with an elastic buffer layer was constructed using the following four steps:

Step 1: Fabrication of the inner spherical shell.

First, as shown in Figure 6a, 20 regular hexagons and 12 regular pentagons were cut from thin metal plates (stainless-steel plates with thicknesses of 1.0 mm were used in this study). Subsequently, the sides of the regular hexagons and pentagons were welded to fabricate a soccer-ball-shaped box, as shown in Figure 6b. As shown in Figure 6c, the box was filled with tap water. As shown in Figure 6d, a manual hydraulic pump was used to apply hydraulic pressure inside the ball-shaped box for plastic forming. Consequently, a spherical, soccer-ball-shaped, thin-walled shell was obtained, as shown in Figure 6e.

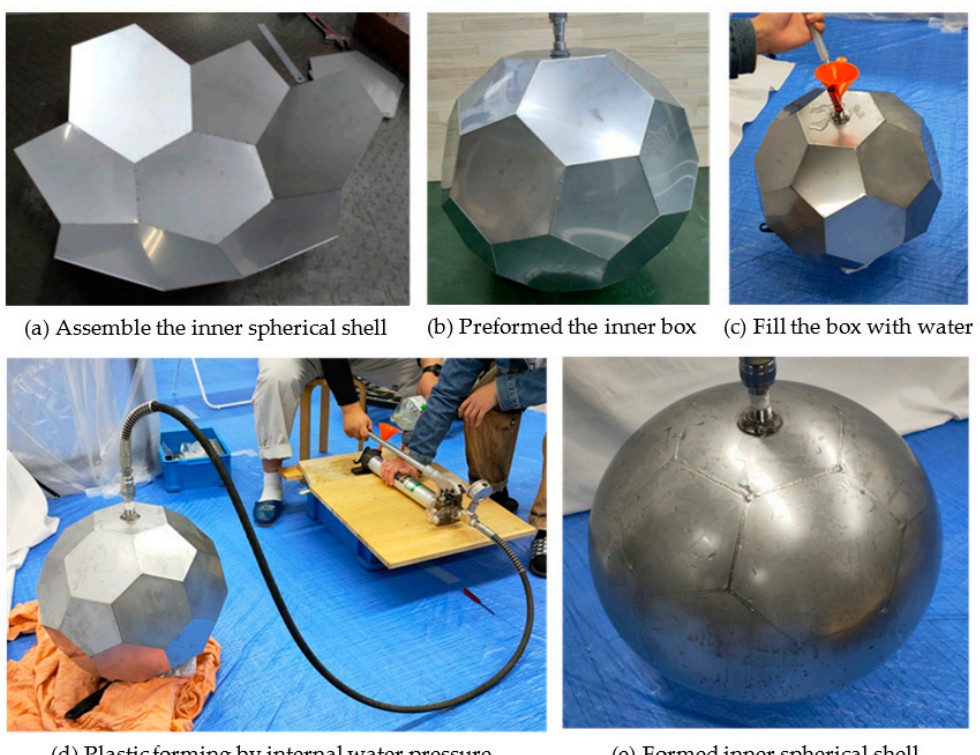

(a) Assemble the inner spherical shell    (b) Preformed the inner box    (c) Fill the box with water

(d) Plastic forming by internal water pressure      (e) Formed inner spherical shell

**Figure 6.** Fabrication of the thin-walled metal shell structure of the inner layer.

Step 2: Processing of closed, preformed outer shell.

Figure 7a shows the inner spherical shell that was bulged in the previous step. As shown in Figure 7b, a circular hole was created around the water inlet, and a small round bar was welded to the intersection of each polygon. Here, round bars with the same length as the assumed elastic buffer layer thickness of 30 mm were used. Subsequently, as shown in Figure 7c–e, new regular hexagonal and pentagonal metal plates were welded to the ends of the small round bars. Finally, a regular hexagonal metal plate was welded with a central bulkhead socket. An outer layer of the closed, preformed shell was obtained, as shown in Figure 7f.

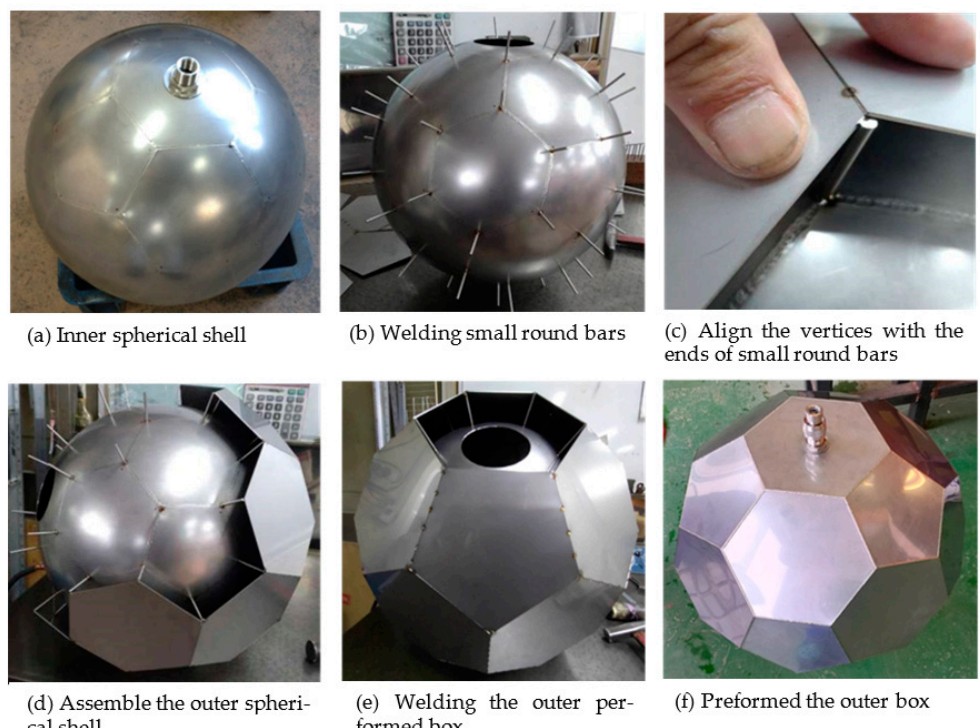

(a) Inner spherical shell

(b) Welding small round bars

(c) Align the vertices with the ends of small round bars

(d) Assemble the outer spherical shell

(e) Welding the outer performed box

(f) Preformed the outer box

**Figure 7.** Fabrication of the outer layer of the closed, preformed shell.

Figure 8 shows a cross-sectional view of the welded-closed, preformed outer shell. The gap between the closed, preformed inner and outer shells was connected to the inner layer of the shell through a circular hole. When water pressure was applied from the outside, the water pressures inside and outside the inner spherical shell were always equal. Therefore, the inner shell did not undergo plastic deformation, and only the closed, preformed outer shell bulge was formed.

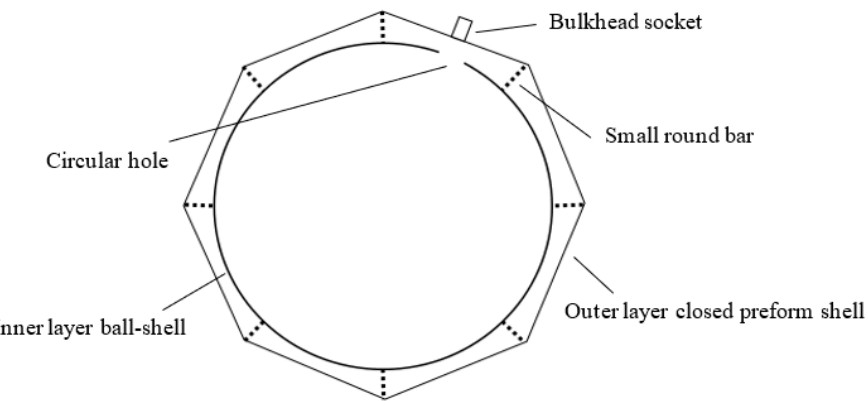

**Figure 8.** Cross-section of the thin-walled spherical structure.

Step 3: Bulge formation of the outer layer of the shell.

A double spherical shell was produced according to the procedure illustrated in Figure 9 for the thin structure shown in Figure 7. First, the interior of the thin-walled structure was filled with water through a bulkhead socket, as shown in Figure 9a. Subsequently, as shown in Figure 9b, a manual hydraulic pump was used to apply water pressure inside the thin-walled structure and bulge it. Consequently, a double spherical shell was obtained, as shown in Figure 9c.

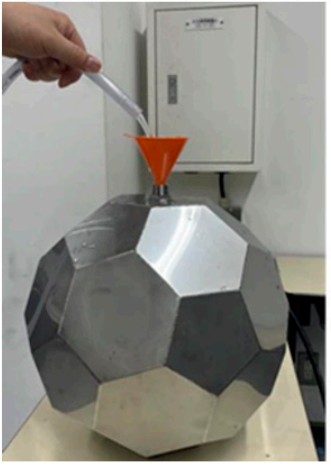 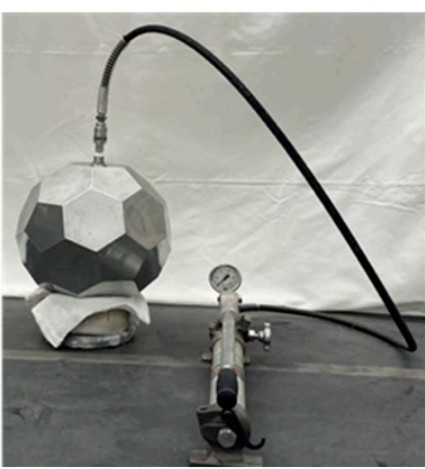 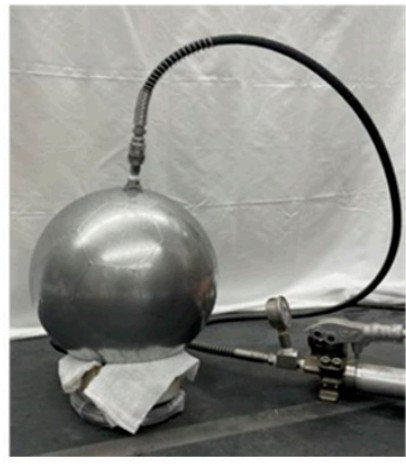

(a) Fill the outer preformed box with water

(b) Plastic forming by internal water pressure

(c) Formed outer spherical shell

**Figure 9.** Fabrication of the outer layer of the thin-walled metallic structure.

Step 4: Fabrication of the intermediate elastic cushioning layer.

Using the double spherical shell shown in Figure 9 as a base, a laminated spherical shell structure with an elastic buffer layer was produced according to the procedure illustrated in Figure 10. As shown in Figure 10a, a circular hole was created around the bulkhead socket of the double spherical shell. Subsequently, as shown in Figure 10b, adhesive tape was used to temporarily fix the circular hole of the inner spherical shell. In addition, silicone rubber and curing agents were mixed at a ratio of 3% and sufficiently stirred, as shown in Figure 10c. Furthermore, as shown in Figure 10d, a mixture of the silicone rubber and curing agents was injected between the inner and outer layers of the double spherical shell. The filled state is shown in Figure 10e. Finally, the silicone-rubber-filled double spherical shell was left for several hours to completely harden the rubber. As shown in Figure 10f, the silicone rubber of the circular hole was removed, and an intermediate elastic buffer layer for the double spherical shell was formed through the circular hole.

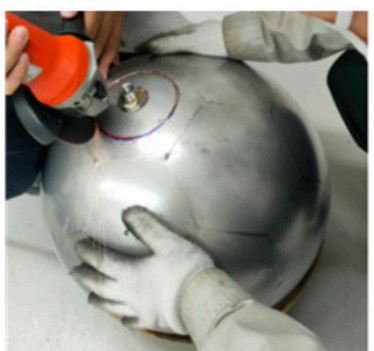 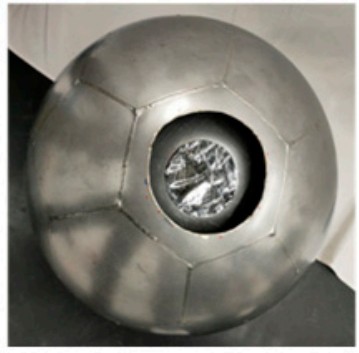

(a) Cut a circular hole

(b) Apply temporary tape to the inner spherical shell hole

**Figure 10.** *Cont.*

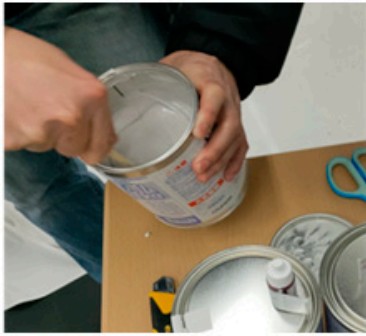
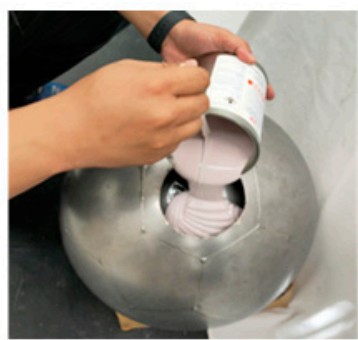
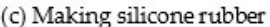

(c) Making silicone rubber

(d) Inject between inner and outer spherical shell

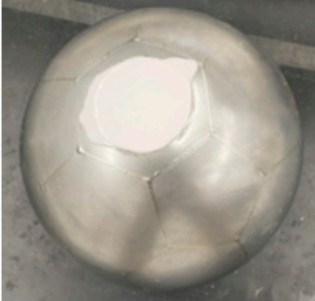
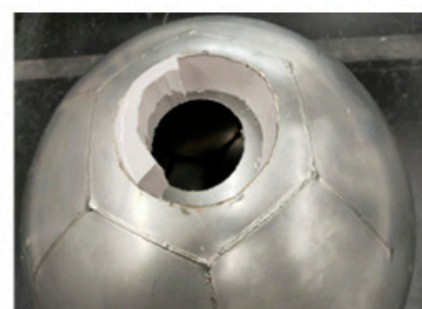
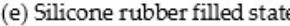

(e) Silicone rubber filled state

(f) Remove silicone rubber around the circular hole

**Figure 10.** Fabrication of the laminated spherical shell structure with an elastic buffer layer.

For practical use, a laminated spherical tsunami shelter with an elastic buffer layer was obtained by modifying the circular hole, which acted as an entrance.

As a future research topic, injecting expanded urethane foam between the double spherical shells will be considered to reduce the weight and improve the cushioning effect of the tsunami shelter, as shown in Figure 10.

## 3. Results and Discussion

### 3.1. Shape Accuracy of the Bulge Formed in the Spherical Shell

To confirm the accuracy of the shape of the fabricated spherical shell, the measurement system shown in Figure 11 was fabricated. The outer dimensions of the fabricated spherical shell were determined using a camera stand, a rotary table, and a laser displacement meter (OPTEX CD22-35VM12; measurement accuracy: ±0.01 mm).

The fabricated spherical shell was fixed on the rotary table, as shown in Figure 11. The rotation angle of the spherical shell was accurately measured by turning the handle of the rotary table.

The height from the pedestal to the laser displacement meter was measured by moving the horizontal beam up and down on the stand. A laser displacement meter was attached to the tip of the horizontal beam on the stand. The distance between the laser displacement meter and the surface of the spherical shell was measured and recorded using a data logger.

As shown in Figure 12, the height of the laser displacement meter, the rotation angle of the rotary table, and the distance from the laser displacement meter were measured, and coordinate conversion was performed. Consequently, the 3D coordinates of the sample points on the surface of the spherical shell were obtained. The spherical shell to be measured had seven cross-sections at intervals of 30 mm in the vertical direction, centered on the central cross-section, L0. For each cross-section, 72 measurement sample points were selected at 5° intervals in the circumferential direction. The center coordinates and average radius were calculated by averaging the coordinate values of each sampling point.

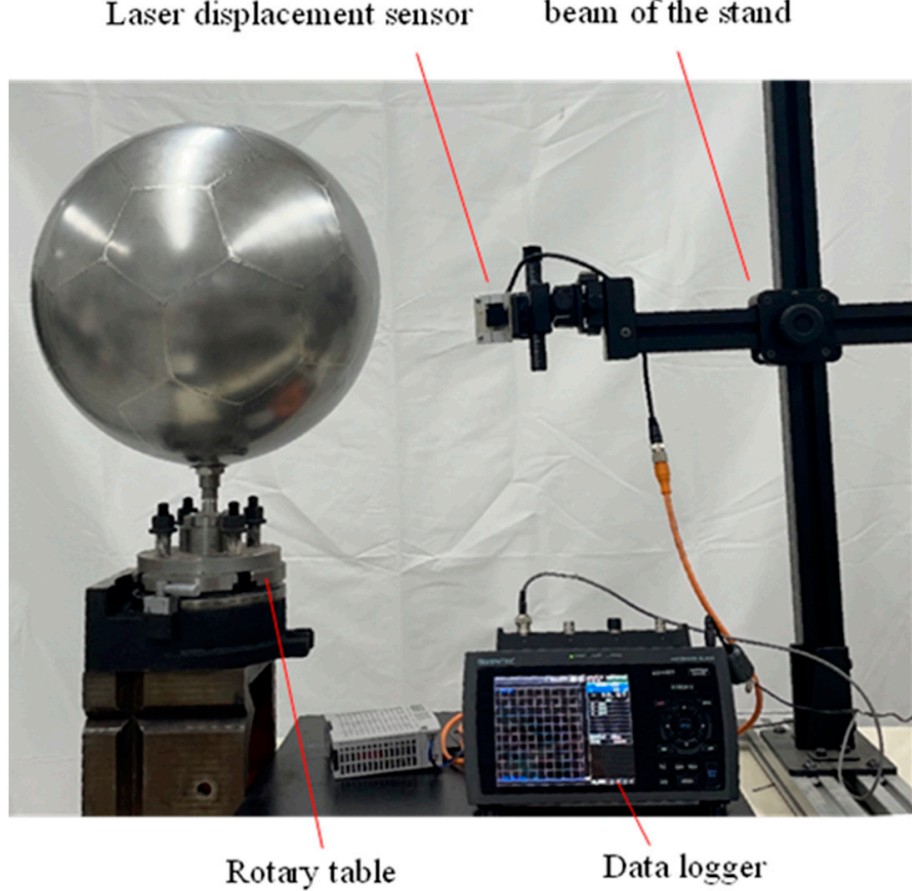

**Figure 11.** Device for measuring the roundness of the formed spherical shell.

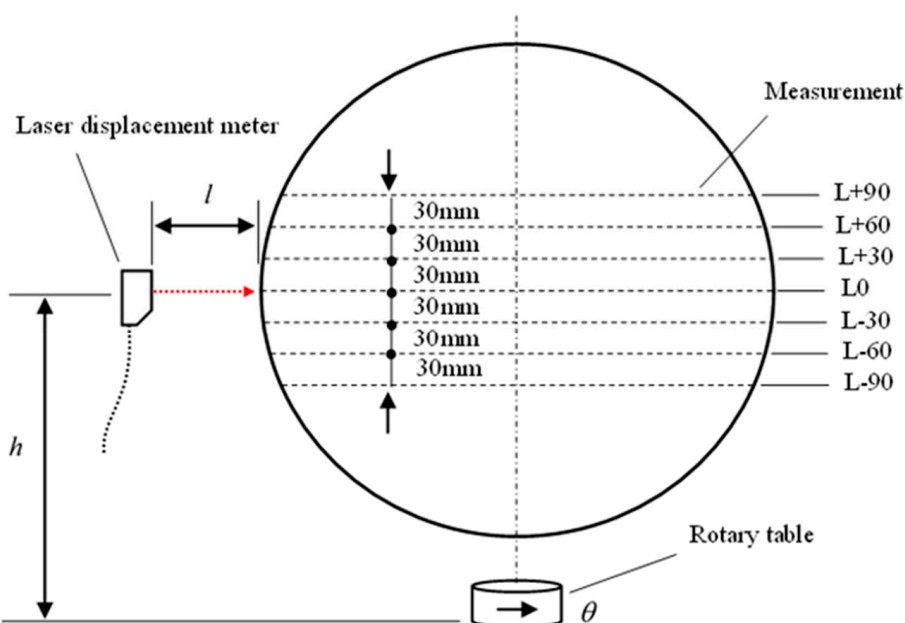

**Figure 12.** Measurement sample points and measurement parameters of the fabricated spherical shell.

Figures 13 and 14 show the measurement results of the sampling points. Each dotted red line indicates the coordinate position of sampling points, and each solid blue line indicates the true circle in the cross-section.

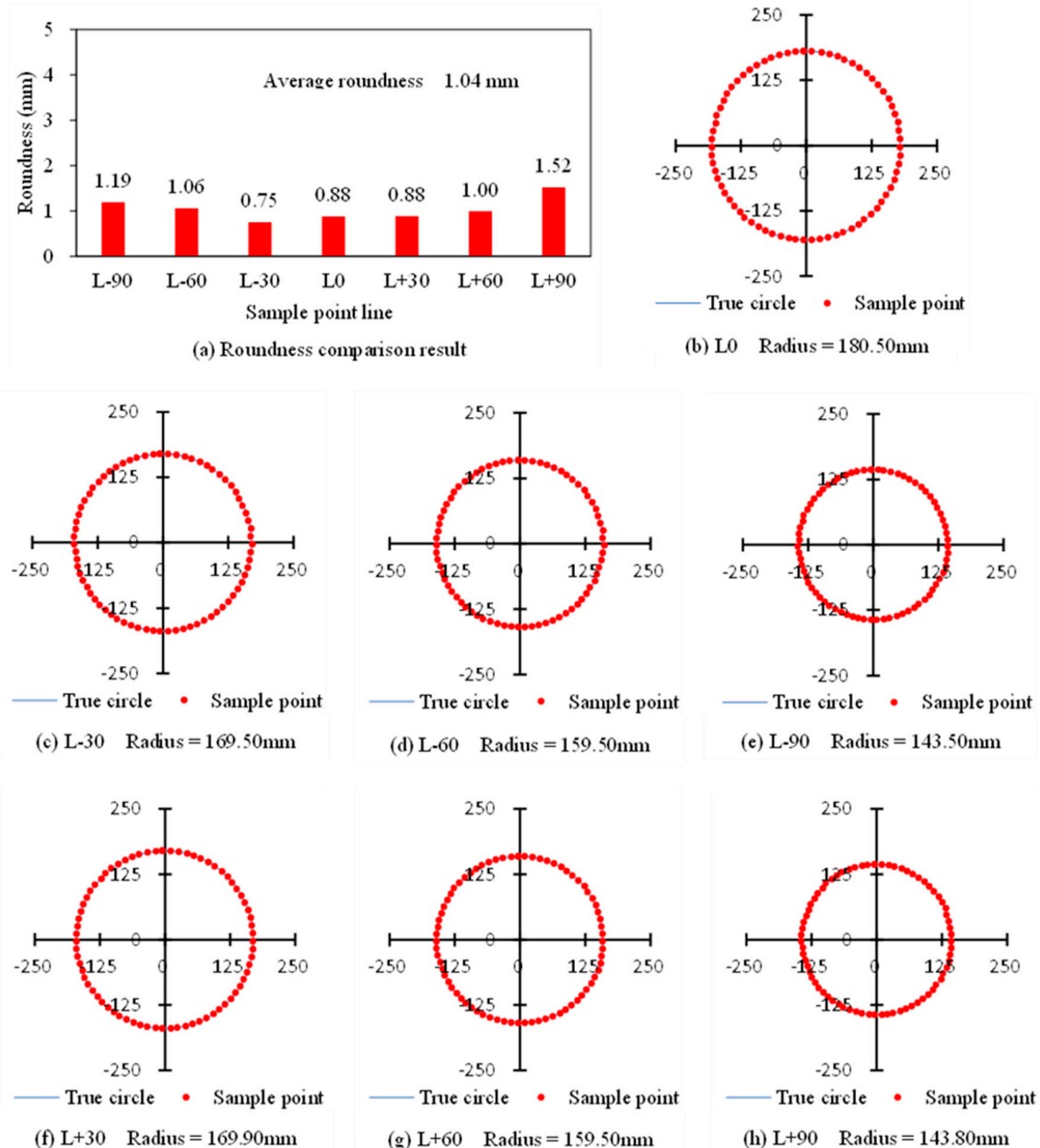

**Figure 13.** Results of measuring the shape of the inner-layer spherical shell.

Figure 13 shows the measurement results for the outer shape of the inner spherical shell processed in the first step. Figure 13a shows the comparison of the roundness of the cross-sections. The maximum and average circularity values were 0.75 and 1.52 mm, respectively, and the average circularity value of the seven cross-sections was 1.04 mm. The roundness of the inner spherical shell, with a diameter of approximately 360 mm, was small; therefore, the machining accuracy was considered good.

Figure 13b shows the measurement results for the central section, L0. The sampling points were uniformly located on the circumference, and the forming quality, represented by the circle, was good, with an average radius of 180.50 mm.

Figure 13c–e show the measurements of three cross-sections selected at 30 mm intervals below the central section of the spherical shell. It can be observed that the sampling points were uniformly located below the circumference. The average radii of cross-sections L-30, L-60, and L-90 were 169.50, 159.50, and 143.50 mm, respectively.

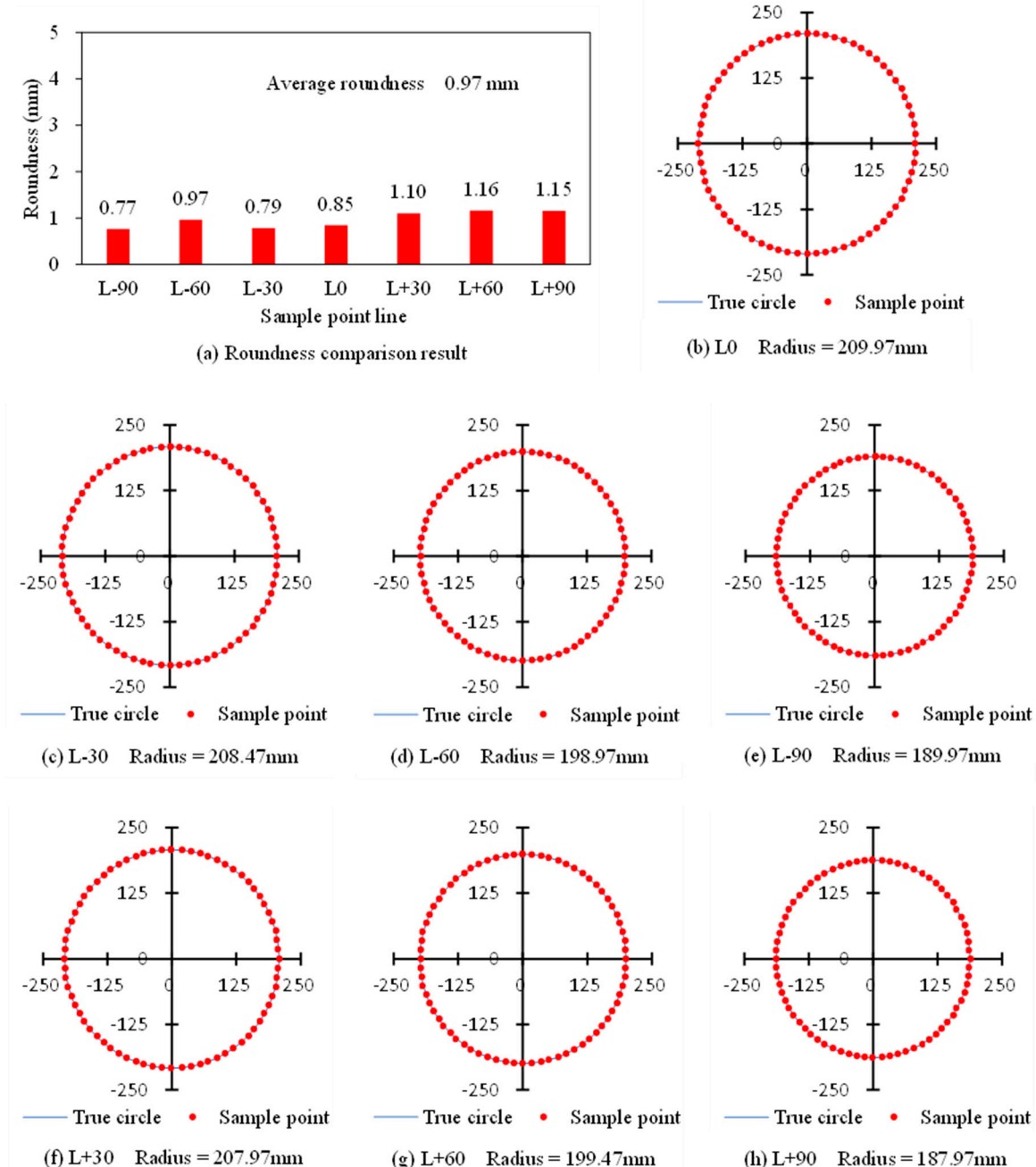

**Figure 14.** Results of measuring the shape of the outer-layer spherical shell.

Figure 13f–h show the measurements of three cross-sections selected above the center of the spherical shell at 30 mm intervals. It can be observed that the sampling points were uniformly positioned on the circumference. The average radii of the circumferences at cross-sections L+30, L+60, and L+90 were 169.90, 159.50, and 143.80 mm, respectively.

Figure 14 shows the measurements of the outer spherical shell, obtained in the first step, and compares the roundness values of the cross-sections. The minimum, maximum, and average circularity values were 0.77, 1.16, and 0.97 mm, respectively. Because the circularity of the outer spherical shell, with a diameter of approximately 420 mm, was small, the machining accuracy was considered good.

Figure 14b shows the measurements of the central section, L0. The sampling points were uniformly located on the circumference, and the forming quality, represented by the circle, was good, with an average radius of 209.97 mm.

Figure 14c–e show the measurements of three cross-sections located from the center to the bottom of the spherical shell at 30 mm intervals. It can be observed that the measured sampling points were uniformly located below the circumference. The average radii of cross-sections L-30, L-60, and L-90 were 208.47, 198.97, and 189.97 mm, respectively.

Figure 14f–h show the measurements of three cross-sections located from the center of the spherical shell upward at 30 mm intervals. It can be observed that the sampling points were uniformly positioned on the circumference. The average radii of cross-sections L+30, L+60, and L+90 were 207.97, 199.47, and 187.97 mm, respectively.

The measurement results shown in Figures 13 and 14 indicate that the accuracy of the shape of the laminated spherical tsunami shelter with buffer layers processed using the proposed method was good.

### 3.2. Calculation of Basic Part Parameters and Water Pressure

When the proposed processing method is practically applied, it will be necessary to determine the side lengths of the regular hexagonal and pentagonal parts based on the design radius of the spherical shell. Design Equation (2) was derived to solve this problem. The validity of Equation (2) was confirmed using the measurement results of the fabricated spherical shell. Table 1 lists the relevant shape parameters of the inner and outer spherical shells.

**Table 1.** Verification of part design formula in formation process of spherical shell.

| Spherical Shell | Polygon Part Side Length $a$ (mm) | Average Radius of Spherical Shell, $R$ (mm) | | |
|---|---|---|---|---|
| | | Design Value | Measured Value | Error |
| Inner | 72 | 180 | 180.50 | 0.28% |
| Outer | 84 | 210 | 209.97 | −0.01% |

As shown in Table 1, the side lengths of the regular hexagonal and pentagonal flat plates for the inner spherical shell were 72 mm each, and the target radius of the designed inner spherical shell was 180 mm. The measured radius of the spherical shell obtained using the IHBF method was 180.50 mm, with an error of 0.28%. The side lengths of the regular hexagonal and pentagonal flat plates for the outer spherical shell were 84 mm, and the target radius of the designed outer spherical shell was 210 mm. The measured radius of the spherical shell obtained using the IHBF method was 209.97 mm, with an error of −0.01%.

Therefore, the design of the basic parts was validated using Equation (2). Because the only parameter for designing regular hexagonal and pentagonal shapes is the side length, Equation (2) was used to design the basic parts of the spherical shell.

Furthermore, to determine the water pressure required for bulging using the IHBF method, as shown in Figure 15, the spherical shell was cut in half, and the following equation was obtained from the equilibrium relationship of the internal force:

$$\sigma = \frac{PR}{2t} \tag{3}$$

where $R$ is the radius of the spherical shell, $t$ is the wall thickness, $\sigma$ is the tensile stress in the side walls, and $\sigma_s$ is the yield stress of SUS304 when plastic deformation occurs. When internal water pressure is applied, the stress acting on the side wall of a spherical shell begins to generate plastic deformation until it reaches the yield stress of the material. The required internal water pressure can be calculated using the following equation:

$$P = \frac{2t\sigma_s}{R} \tag{4}$$

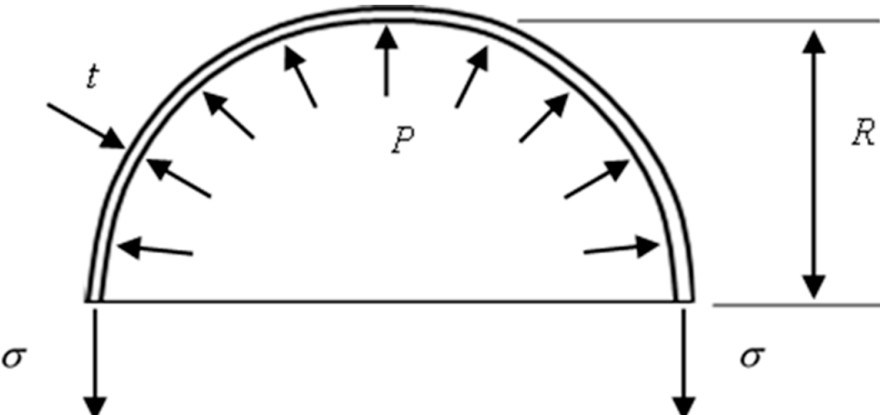

**Figure 15.** Internal hydraulic pressure and tensile stress during bulging formation.

In Formulas (5) and (6), $P_{in}$ and $P_{out}$, respectively, represent the theoretical water pressure values required of the inner and the outer spherical shell using the IHBF method.

In the experimental example in this study, the yield stress of the stainless-steel SUS 304 plate was 255 MPa, the wall thickness was 1.0 mm, the radius of the inner spherical shell was 180 mm, and the radius of the outer spherical shell was 210 mm. By substituting these parameters into Equation (4), the water pressure at which plastic deformation begins was calculated using the following equations:

$$P_{in} = \frac{2 \times 0.001 \times 255}{0.18} = 2.83 \text{ MPa} \tag{5}$$

$$P_{out} = \frac{2 \times 0.001 \times 255}{0.21} = 2.43 \text{ MPa} \tag{6}$$

In the actual bulging experiment, the maximum pressures read from the meter of the manual hydraulic pump were 3.1 and 2.7 MPa, which were 9.5% and 10.0% higher, respectively, than the calculated values obtained using Equations (5) and (6), respectively. This could be attributed to the effect of work hardening the wall material during the bulge-forming process.

### 3.3. Cushioning Effect of Laminated Spherical Shell

To verify the cushioning effect of the proposed laminated spherical tsunami shelter, a hammer impact test was conducted, as shown in Figure 16. The acceleration signals from when the spherical tsunami shelter was hit with a hammer were recorded using acceleration sensors attached to the inner and outer sides of the shelter, and the difference in acceleration was used to evaluate the buffering effect of the shelter. Because the measured acceleration signals had a random distribution, the acceleration standard deviation, $S_a$, was used to evaluate the degree of amplitude fluctuation, expressed with the following equation:

$$S_a = \sqrt{\frac{1}{N} \sum_{i=1}^{N} (a_i - a_{aver})^2} \tag{7}$$

where $a_i$ is the measured value of the acceleration, $a_{aver}$ is the average value of the measured acceleration, and $N$ is the total number of measurement samples in the impact test.

Figure 17a shows the fabricated impact test device and measurement system, which consisted of a bulge-forming laminated spherical tsunami shelter, a striking hammer, two accelerometers, an FFT analyzer, and a test computer.

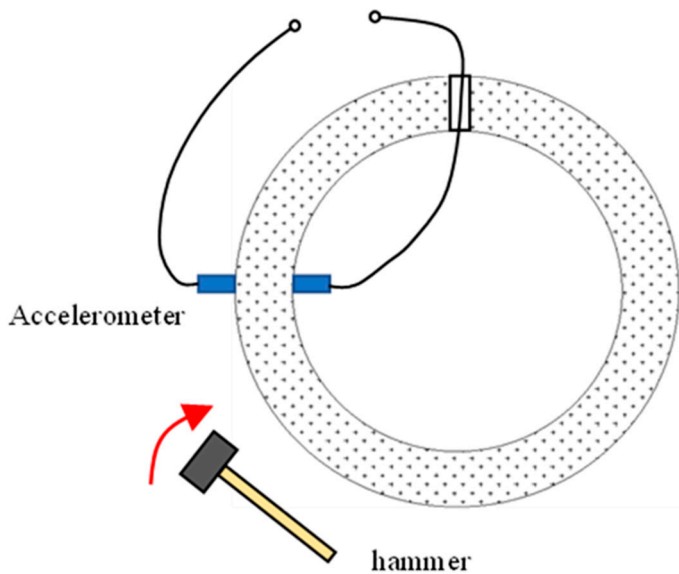

**Figure 16.** Schematic of the impact test to verify the cushioning effect of the spherical ball.

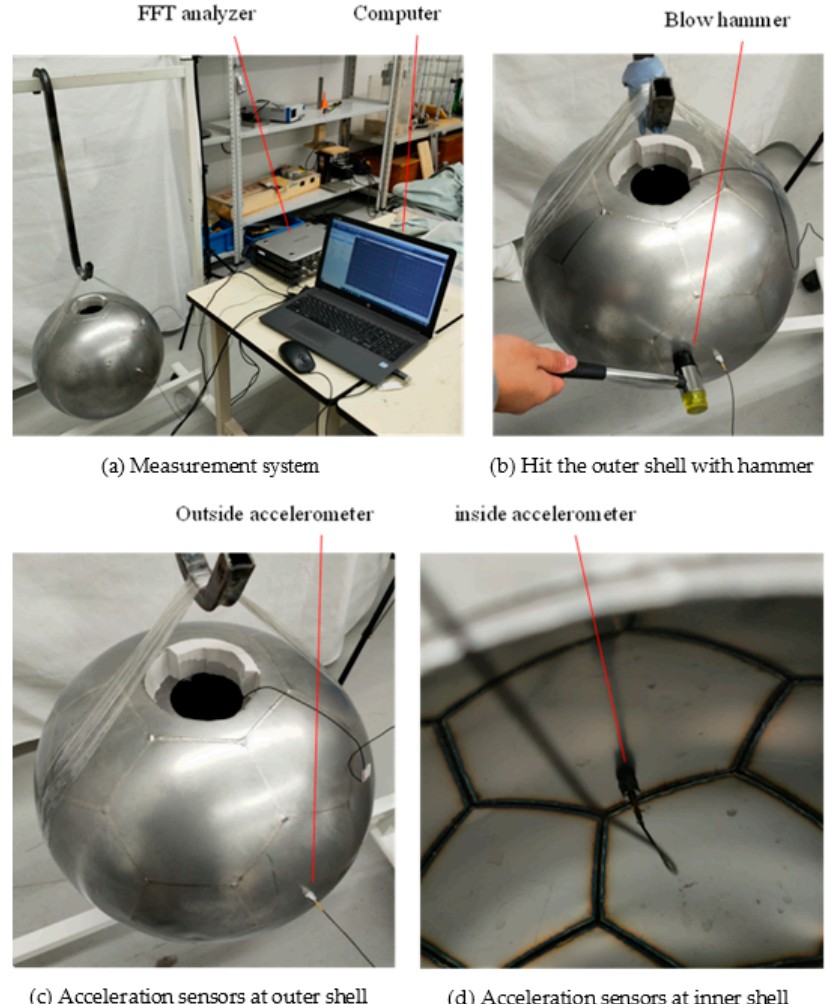

**Figure 17.** Impact test verifying the cushioning effect of the spherical shell.

To measure the cushioning effect, the accelerometers were affixed to corresponding locations on the outer and inner surfaces. The hammer was hit as close as possible to the accelerometer, as shown in Figure 17b. Figure 17c,d show the acceleration sensors attached to the outer and inner surfaces, respectively.

Figure 18 shows the measurement results obtained from the accelerometers during an actual impact test. The solid red line represents the acceleration results for the outer surface, and the blue dashed line represents the acceleration results for the inner surface.

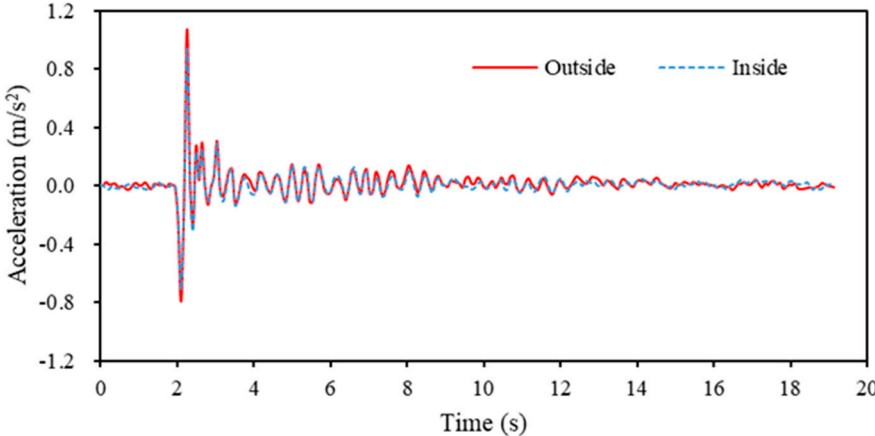

**Figure 18.** Measurement result of acceleration of inner and outer layers of spherical ball shell.

The maximum radial accelerations on the outer and inner surfaces were 1.07 and 0.94 m/s$^2$, respectively. In comparing both values, it was found that the maximum acceleration of the inner surface was reduced by 14.16%.

The standard deviation values of the acceleration in the outer and inner surfaces were 0.10 and 0.09 m/s$^2$, respectively. In comparing both values, it was found that the standard deviation of the acceleration in the inner surface was reduced by 10.17%.

Therefore, it was confirmed that the cushioning effect of the laminated spherical tsunami shelter was more than 10%.

Figure 19 shows the comparison of the power spectral densities of the shock response acceleration measured on the outer and inner surfaces of the laminated spherical tsunami shelter. The red line is the result for the outer surface and the blue dashed line is for the inner surface. Figure 19 shows that the acceleration power spectral density value is remarkably large around the frequency of 2.66 Hz, indicating that the main frequency component of the impact hammer load is 2.66 Hz. It can also be seen throughout the graph that the power spectral density values for the inner surface are relatively smaller than for the outer surface.

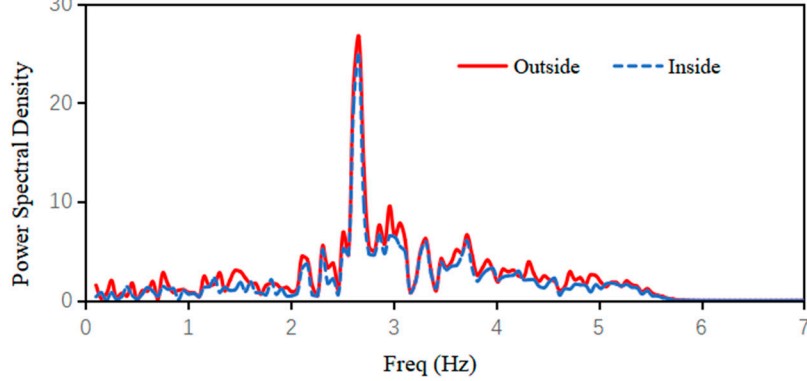

**Figure 19.** Power spectral density results of acceleration of inner and outer layers of spherical ball shell.

However, the cushioning effect varies considerably depending on the type of cushioning material between the inner and outer layers. In this study, silicone rubber was used as the cushioning material. However, for future studies, it is expected that the application of new cushioning materials, such as urethane foam, will improve the cushioning effect.

## 4. Conclusions

In this study, we developed a laminated spherical tsunami shelter with an elastic buffer layer using the IHBF method. The following conclusions were drawn from the bulge-forming experiment and impact test performed on the laminated spherical tsunami shelter.

(1) The proposed processing method only requires the cutting of regular hexagonal and regular pentagonal parts from a flat steel plate, eliminating the need for the conventional press formation process for bending flat plates. Welding the sides of the hexagons and pentagons resulted in a geometrically unique soccer-ball shape; therefore, conventional jigs for determining welding positions were omitted. It was confirmed that this could be processed using a simple method.

(2) The experimental results confirmed that the shape accuracy of the fabricated spherical shell was high because the internal water pressure was completely symmetrical. When the design radii of the spherical shells were 180 and 210 mm, respectively, the measured circularity values of the respective inner and outer spherical shells were 0.88 and 0.85 mm, respectively.

(3) To put this processing method into practical use, we derived a design formula, $a = 0.40355R$, for regular hexagonal and pentagonal plates to form the basic parts. As a result of the forming experiments, parts with side lengths of 72 and 84 mm were used; the design radii of the inner and outer spherical ball shells were 180 and 210 mm, respectively. The measured radii of the actual bulge-formed inner and outer spherical shells were 180.50 and 209.97 mm, respectively, and the errors from the design radii were 0.28% and −0.01%, respectively. The design equations for the basic parts derived in this study were verified to be accurate.

(4) A formula for calculating the internal water pressure required for actual bulge formation was derived. The results verified by the bulge-forming experiment confirmed that the results of the derived hydraulic formula were similar to the experimental values.

(5) In order to confirm the buffering effect of the laminated spherical tsunami shelter that was processed using the proposed method, we performed a hammering test and confirmed that the buffering effect was greater than 10%. However, the cushioning effect varied considerably, depending on the type of cushioning material between the inner and outer layers. In this study, silicone rubber was used as the cushioning material. However, for future studies, it is expected that the application of new cushioning materials, such as urethane foam, will improve the cushioning effect.

In this study, a scaled-down model of a commercial spherical tsunami shelter was examined with a ratio of about 1:3. Notwithstanding, when developing a larger laminated spherical tsunami shelter in the future, it will be necessary to conduct an integrated study that will include not only the processing method but also the riding comfort of the evacuees inside.

**Author Contributions:** Writing—original draft preparation, J.H.; writing—review and editing, J.G. and X.Z.; data curation, L.C.; investigation, I.H. and W.Z.; software, L.C.; conceptualization, W.Z. and X.Z.; methodology, J.G. and J.H.; validation, I.H. All authors have read and agreed to the published version of the manuscript.

**Funding:** This research received no external funding.

**Data Availability Statement:** Data are available in a publicly accessible repository.

**Conflicts of Interest:** The authors declare no conflict of interest.

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
