# Peer review of "A Laminated Spherical Tsunami Shelter with an Elastic Buffer Layer and Its Integrated Bulge Processing Method"

_designs, 2023_

Round 1

Reviewer 1 Report

This paper presents an innovative three-layer laminated spherical tsunami shelter and proposes it as an alternative to the conventional FRP thin-walled spherical tsunami shelter. A hammer impact load was applied to the outer layer, and the response acceleration values measured by the acceleration sensors in the inner and outer layers were compared. It was verified that the spherical tsunami shelter proposed in this study has a good cushioning effect and processing performance.

This solution proposed isolator is interesting. Nonetheless, the lack of experimental measurements needs to be clarified before considering this paper for publication.

Please find the suggestions below:

  1. Please specify if this shelter is a lab-scaled model and what the limitations might be in manufacturing larger shelters.
  2. The state-of-the-art considered only a few studies related to thin spherical structures under external load. Please refer to the following papers for a clear system assessment:
    • Li, S., Wang, Z., Wu, G., Zhao, L., & Li, X. (2014). Dynamic response of a sandwich spherical shell with graded metallic foam cores subjected to blast loading. Composites Part A: Applied Science and Manufacturing, 56, 262-271.
    • Iarriccio G, Pellicano F. Nonlinear Dynamics and Stability of Shallow Spherical Caps Under Pressure Loading. J Comput Nonlinear Dyn 2021; 16: 1–8. DOI:10.1115/1.4049080.
  3. Line 133: Please correct the angle definition from "oo6m" to "oO6m."
  4. Table 1: Please specify that you are referring to the average measured radius and clarify that the error is also averaged.
  5. Formula (3) and (4): Insert the symbols σ and σs close to their definitions within the text, similar to how it was done for the radius R and the thickness t.
  6. Formulas (5) and (6): Please specify the meaning of Pin and Pout.
  7. Lines 301-303 and 312-316: Please consider removing the redundancy regarding the 10% pressure approximation to improve readability.
  8. Line 343: Please consider replacing "vertical acceleration" with "radial acceleration" as the latter seems to better describe the presented experimental measurements.
  9. Figure 18: The caption is not related to the given figure.
  10. Please explain why the frequency response function (FRF) or other spectral quantities like the power spectral density were not considered for evaluating the cushioning effect. The acceleration difference, to my knowledge, is not the best method. Moreover, experiments were conducted by hitting the structure with a non-instrumented hammer, which was not intended for impact testing. This detrimentally affects the validity of your analysis.
  11. Figure 18 discussion lacks details, and it is not specified where formula (7) was used.

Reviewer 2 Report

Authors in their paper on “Laminated Spherical Tsunami Shelter with Elastic Buffer Layer 2 and Its Integrated Bulge Processing Method” proposed a novel three-layer laminated spherical tsunami shelter and its fabrication method as an alternative to the conventional FRP thin-walled spherical tsunami shelter.  The inner and outer layers were made of thin-walled stainless-steel spherical shells using the integral hydro-bulge forming (IHBF) method. The inter-layers between the inner and outer layers were filled with elastic rubber. Acceleration sensors were attached to the inner and outer layers of the processed laminated spherical tsunami shelter. The work is of interest to the profession and it can be accepted.

It is good.

Reviewer 3 Report

The research presents a novel three-layer laminated spherical tsunami shelter and its fabrication method. The experimental results confirmed that the shape accuracy of the fabricated spherical shell was high. The spherical tsunami shelter was experimentally verified to have a good cushioning effect and processing performance.

However, the following questions and comments need to be considered before the paper can be accepted for publication.

1. Please explain why we should use double spheres instead of using double icosahedra directly, which can avoid the complicated production process such as pressurization.

2. Please describe in detail how the water inlet in Figure 6(c) is connected to the surrounding and how it is sealed.

3. How the length of the small round bars in Figure 7(b) was determined. Does its length affect the serviceability of spherical tsunami shelter?

4. Line 302 mentioned “the water pressure required for actual bulge formation should be set at approximately 10 % higher than the value at the start of plastic deformation”. Could the author explain how this 10% estimate was obtained?

5. Some recent and related studies on cubic polyhedral and/or origami structures can be briefly mentioned. Please refer to: A class of expandable polyhedral structures; Multi-stability of hexagonal origami hypar based on group theory and symmetry breaking; Selective hinge removal strategy for architecting hierarchical auxetic metamaterials; Two-orbit switch-pitch structures; Morphing of curved corrugated shells.

Round 2

Reviewer 1 Report

The authors have addressed all of my concerns with the original manuscript.

The revised manuscript is now ready for publication.